# PROM SSCOL—Impact of a Papillomavirus Vaccination Promotion Program in Middle Schools to Raise the Vaccinal Coverage on Reunion Island

**DOI:** 10.3390/vaccines10111923

**Published:** 2022-11-14

**Authors:** Phuong Lien Tran, Emmanuel Chirpaz, Malik Boukerrou, Antoine Bertolotti

**Affiliations:** 1Service de Gynécologie et Obstétrique, Centre Hospitalier Universitaire (CHU) Sud La Réunion, Avenue du Président Mitterrand, BP350, 97448 Saint Pierre, La Réunion, France; 2Centre d’Etudes Périnatales de l’Océan Indien, Centre Hospitalier Universitaire (CHU) Sud Réunion, Avenue du Président Mitterrand, BP350, 97448 Saint Pierre, La Réunion, France; 3Cancer Registry of Reunion Island, Centre Hospitalier Universitaire (CHU) Nord Réunion, Felix Guyon, Allée des Topazes CS 11 021, 97400 Saint-Denis, La Réunion, France; 4INSERM, CIC 1410, Centre Hospitalier Universitaire (CHU) Sud Réunion, Avenue du Président Mitterrand, BP350, 97448 Saint Pierre, La Réunion, France; 5Service des Maladies Infectieuses-Dermatologie, Centre Hospitalier Universitaire (CHU) Sud Réunion, Avenue du Président Mitterrand, BP350, 97448 Saint Pierre, La Réunion, France

**Keywords:** HPV vaccination, middle school, health program

## Abstract

Introduction: On Reunion Island, cervical cancer is the third most common cause of cancer in women. Primary prevention is based on the HPV vaccination, yet coverage rate is low (8.1%). The objective of the study was to evaluate the impact of a health promotion program on the proportion of middle school girls who have completed the HPV vaccination schedule. Material and methods: In this prospective, controlled intervention study of superiority, 12 classes were randomly selected in an intervention school where the promotion program took place, and in a control school where no specific intervention was planned. The program combined: information to students during school classes, information to parents by letter and phone calls, information to general practitioners by letter and video conference call, and the free school-based vaccination (in a “health bus” parked in the schoolyard) with the nonavalent HPV vaccine. Results: In the intervention group, the completion was achieved for 26 girls, which was significantly higher than in the control group (three girls, *p* < 10^−3^). The initiated vaccination was also higher in the intervention group (31 girls vs. 6 girls in the control group, *p* < 10^−3^). The same results were obtained for the boys as for the full or partial scheme (seven boys vs. 0, *p* = 0.01; 16 boys vs. 1, *p* < 10^−3^, respectively). Conclusions: Implementing a health promotion program and offering the free, school-based vaccination raised the vaccination coverage. These results are promising and may be a stepping stone to expanding this program to the whole Reunion Island and hopefully someday decrease the burden of cervical cancer.

## 1. Introduction

On Reunion Island (a French overseas territory located near the eastern coast of Madagascar, in the Indian Ocean), uterine cervical cancer is the fourth most common cause of cancer in women. The standardized mortality rate accounts for 4.8 in 100,000 women, three times higher than on mainland France [1,2].

CC results from the human papillomavirus (HPV) infection. The primary prevention is based on the HPV vaccination, which has proven to be effective in reducing the prevalence of HPV carriage, and the incidence of condyloma or intermediate grade cervical dysplasia [3,4], and invasive CC [5]. Thus, in Australia, where the HPV vaccination coverage is high, the incidence rate of CC could decline to less than 1/100,000 PY, by 2066 [6].

In France, before December 2019, the HPV vaccination was recommended for girls only. Since HPV infections can also lead to vulvar, vaginal, penile, anal or oropharyngeal cancers, France now recommends a gender-neutral vaccination, in order to promote herd immunity and reduce the circulation of the virus in the general population [3,7]. Because HPV is mainly transmitted sexually, it is important to vaccinate at young age. Vaccination is recommended for adolescents aged 11 to 14 years (two doses), with a catch-up vaccination possible between 15 and 19 years of age (three doses). High levels of vaccination coverage are obtained in countries that vaccinate in schools [8,9,10].

On Reunion Island, the HPV vaccination coverage rate is the lowest in France, estimated at 8.1% among 16-year-old girls in 2018, while the national average is already low (23.7%) [11]; yet 96.8% of the genotypes circulating on the Island are covered by the nonavalent vaccine [12]. Through questionnaires to mothers and general practitioners (GPs), it appeared that a lack of information and the vaccination mistrust from parents, as well as from GPs, are the main factors contributing to the low HPV vaccination coverage [13]. Inhabitants were poorly informed about the existence of this vaccine [13], reinforced by general practitioners’ doubts in whom patients trust [14]. Moreover, the vaccination coverage rates depend on the socioeconomic level of the population [15], and Reunion Island is one of the poorest French departments [16]. Therefore, given the epidemiological situation in Reunion Island (a high incidence and mortality for CC, a very low coverage rate for the HPV vaccination), interventions targeting health professionals appear to be paramount, especially when combined with interventions targeting the population to be vaccinated and their parents [17].

We conducted our study soon after the French recommendations for the gender neutral HPV vaccination, and since the acceptability of the vaccination in boys had not yet been explored, we decided to evaluate girls and boys, separately.

Thus the main objective of the study was to evaluate, in a population of middle school girls on Reunion Island, the impact of a health promotion program, on the proportion of middle school girls who have completed the full HPV vaccination schedule (two or three doses), by the end of the school year.

## 2. Material and Methods

### 2.1. Setting and Study Design

The full protocol of this prospective, controlled intervention study of superiority is available [18].

The two groups (intervention group and control group) were selected from two middle schools in Southern Reunion Island located in the priority education zone (which theoretically concerns populations with a low socio economic status).

In order to have the most comparable populations in the two arms, we carried out a cluster trial. In each of the schools, we randomly drew three classes in each grade level (6th, 7th, 8th, and 9th grade) to have a balanced number of students in each arm. Thus, 12 classes were selected from each school.

In the intervention group, a health promotion program was conducted during the 2020–2021 academic year (October 2020 to June 2021), combining information to students during school classes, information to parents by letter and phone calls, information to general practitioners by letter and video conference call, and a free school-based vaccination (in a “health bus”) with the nonavalent HPV vaccine.

Children in the selected classes were asked to bring in their health record on a specific date. On that day, an investigator collected the data in the health records. During this time, an information session about the anatomy of the genital organs, sexually transmitted diseases and vaccination was given in class, lasting approximately one hour and adapted to the level of understanding (according to grade and age), in partnership with teachers. Health records were immediately returned to the students.

Each child was given an envelope to take home, with a consent form to participate in the study to be signed (by the child him/herself and both parents or holders of parental authority). There was also an information letter, explaining the benefits of the HPV vaccination. Since assemblies and parental meetings were forbidden during the COVID-19 pandemic, parents were phoned and oral consent was collected. During the phone calls, oral information about the HPV vaccination was delivered.

In the control group, no specific intervention was planned.

### 2.2. Objectives

The main objective of the study was to evaluate, in a population of middle school girls on Reunion Island, the impact of a health promotion program, on the proportion of middle school girls who have completed the full HPV vaccination schedule (two or three doses) by the end of the school year.

Secondary objectives in the study population at the end of school year were:(1)to assess the impact of the combined health promotion program on the proportion of middle school girls who initiated the HPV vaccination (at least one dose),(2)to assess the impact of this program among middle school boys (full schedule and initiated vaccination),(3)to assess the acceptability of the HPV vaccination in a school setting.

### 2.3. Data Collection

The data were collected after consent was signed by parents or holders of parental authority, and by the students. The data were collected in the form of self-questionnaires (parents and children), evaluating the socioeconomical data, and prior knowledge and acceptance of the HPV vaccination. The data concerning the vaccination status at the inclusion and at the end of the study were checked in the health record by the investigation team. The data were collected in paper format and were then entered into an electronic case report form (Ennov Clinical) by a clinical study technician.

### 2.4. Statistical Analysis

The qualitative variables were expressed as numbers and percentages with their 95% confidence interval, the quantitative variables as mean with their standard deviation (SD) or median with the 25th and 75th percentile. For the qualitative data, the intervention and control groups were compared by the Chi^2^ test or Fisher’s exact test, according to the validity conditions. Comparisons of the continuous variables were performed by Student’s *t* test or the Wilcoxon test, as appropriate. The analyses were performed on an intention-to-treat basis. The hypotheses were tested with an alpha risk of 0.05, and the confidence intervals were calculated at 95%. All statistical analyses were performed using STATA SE V16^®^ software (StataCorp, College Station, TX, USA).

### 2.5. Ethical Considerations

This research has received the favorable opinion of the research ethics committee (Comité de Protection des personnes (CPP); ethics committee for the protection of individuals) of Ouest II of Angers (No. 20.05.14.35227; 2020/46) and the authorization of the Agence nationale de la sécurité du médicament (ANSM).

## 3. Results

As exposed in the flow chart in Figure 1, the twelve classes randomly selected in each school concerned 245 students in the intervention group (108 girls with a mean age of 12.2 years (CI95%: [11.8; 12.6]) and 137 boys with a mean age of 12.3 years (CI95%: [11.9; 12.7]) and 259 in the control group (125 girls with a mean age of 12.2 years (CI95%: [11.8; 12.7]) and 134 boys with a mean age of 11.9 years (CI95%: [11.5; 12.3])). The repartition among grades 6 to 9 was similar in both groups (*p* = 0.8). The health information, including vaccine status, could be collected for 36.3% (89/245) of the students in the intervention group and 33.6% (87/259) students in the control group (*p* = 0.5). For students for whom the medical information were available, both groups were not significantly different, in terms of age, gender, grade and vaccination status for vaccinations other than HPV (Table 1). 

Of 245 children in the intervention group, some only had one holder of parental authority. For those who had both parents, we tried to call both parents, but when one of them was not available, we considered one oral consent was sufficient. However, both parents’ written consent was necessary. Among 199 mothers who picked up the phone, 79 gave their oral consent for their child’s vaccination. Among the fathers, 39 gave their oral consent for the vaccination among 113 who answered the phone.

A full vaccination was achieved for 24.1% of the 108 girls included in the intervention group (26 girls) by the end of the school year, which was significantly higher than the 2.4% in the control group (three girls; *p* < 10^−3^). When comparing the students with at least one HPV vaccinal dose (initiated vaccination), a higher rate was also obtained in the intervention group (31 girls = 28.7%) than in the control group (six girls = 4.8%; *p* < 10^−3^). Similar results were obtained for boys: full vaccination or initiated vaccination rates were superior in the intervention group (5.1% vs. 0%, *p* = 0.01, and 11.7% vs. 0.7%, *p* < 10^−3^, respectively).

HPV vaccination was initiated for a total of 47 (19.2%) students in the intervention group and 7 (2.7%) in the control group (*p* < 10^−3^). Among the 47 vaccinated children in the intervention group, 37 (78.7%) received at least one dose in the health bus that was parked in their school. Of the 33 fully vaccinated students, 26 (78.8%) received their whole scheme vaccination in the health bus. Of these children vaccinated in the bus, three adverse reactions were reported: one vagal discomfort, two with pain at the injection site, including one who felt dizziness and one who had swelling on the injection site. Concerning the 14 children with an incomplete vaccination at the end of the follow up, eight had initiated their vaccination too late, in regards to the end date of the study in order to have a complete scheme.

At the beginning of the study, 120 general practitioners located around the intervention middle school were sent an information leaflet about the HPV vaccination and were invited to participate in a video conference call about HPV. Seven health professionals participated to the conference.

## 4. Discussion

In a population of middle school students, we implemented a health promotion program during one school year, combining students, parents and general practitioners and provided them with information about the HPV vaccination and the free school-based vaccination in a “health bus”. This intervention significantly increased the HPV vaccination coverage (full vaccination or first dose) in both girls and boys, compared to a control school.

Previous studies have already shown the benefits of school-based educational sessions to improve adolescents’ knowledge and thus their behavior regarding the HPV prevention and to increase the likelihood of the students to become vaccinated [19,20]. Education interventions represent a simple yet potentially effective strategy for increasing the HPV vaccination, especially when targeting groups influential to the HPV vaccination behaviors of adolescents: parents [21], school staff [22] and health care professional [23]. Indeed, knowledge was associated with the recommendation intention and behavior.

In similar studies in the literature, the baseline initiation and completion of the HPV vaccination rates were higher than in our population (baseline: 2.9% on Reunion Island vs. 46.7–93.9% among girls in Canada [24], 76% in New York [25], 16.1% in Sao Paulo [26]). However, the percentage increase points were very heterogenous among the studies (16.3% in Reunion, 2.9% in New York, 34.4% in Sao Paulo). Thus, the impact of these interventions appears to be greater when the baseline vaccination coverage is low.

Most vaccinations identified during the school year, were conducted in the health bus in the intervention school. In other studies, the delivery of the program occurred twice a year to provide both doses in local schools, or nurses went into schools three times a year to deliver the doses. It was perceived that offering the vaccine in schools increased accessibility and convenience [27].

The HPV vaccination coverage achieved at the end of our study was lower than expected, but among a population known to particularly mistrust the HPV vaccination [13], these results may raise hopes for the future, slowly leading to a snowball process. Indeed, children from other classes came nearby the bus during break time, seeking for information about the HPV vaccination, and 30 prescriptions were delivered for these children. Thus, the nearest pharmacy from the intervention school sold 52 HPV doses to 44 children between October 2020 and June 2021, which was two times higher than during the same period the previous year.

This reluctance to the vaccination in France was once more highlighted during the coronavirus pandemic [28]: a high vaccination coverage was slower to achieve than in more compliant Asian countries.

### Strengths and Limits

The strength of this study is its methodology with the randomized selection of school classes and the collection of both parental consent.

Even though our results were significant, the main limitation was the low number of participants, whether it be students (an overall participation rate of 34.9%), or health professionals invited to attend the conference (7/120). This low adherence in the conference could reflect a lack of interest and thus of the prescription of the HPV vaccination. Yet, as mentioned earlier, patients have a deep trust in their general practitioner and the doctor needs to be convinced himself in order to convince the patient and parents. Improvements are needed to sensitize healthcare professionals.

During the study, we were confronted with several difficulties: (i) we encountered great difficulties in obtaining the signed parental consents for the children’s vaccination. Indeed, each child was given an envelope containing information and formalities related to the study, socio-economic questionnaires, etc. [18]. This was probably difficult to understand in a population where illiteracy reaches 23% (two times higher than in Metropolitan France) [29]. However, these logistical barriers, including getting the consent forms returned, the competing priorities within the school setting, have already been identified [27]. Of note, in a similar study conducted in Brazil, when the HPV school base vaccination was implemented, the coverage of the first dose increased from 16.1% to 50.5% (*p* < 0.0001). Nonetheless, according to the Brazilian legislation for the vaccination of children, the vaccines on the National Program of Immunization list do not need the authorization from parents or guardians [26]. (ii) The COVID-19 pandemic, with the impossibility to organize in person meetings with parents and general practitioners, to convince them about the benefits of the vaccine. Moreover, it created a confusing environment between Covid and the HPV vaccine.

Qualitative interviews were conducted to evaluate the barriers to the HPV vaccination in this population, yet sensitized all along school year about its benefits (personal data).

It would have been interesting to evaluate each action separately (information VS free vaccination at school VS control), even though those actions are complementary.

Even though unlikely, students could have accepted to be vaccinated because they wanted to help the researchers and not because they were motivated by the information provided. On the contrary, the wide time offered between the first and last vaccination campaign (December to June), could have been a cause of a low adherence to the second dose. The external validity is needed by extending this program throughout Reunion Island. Since the World Health Organization seems to say that one vaccinal dose may be sufficient, maybe only one-single vaccination campaign may be required.

## 5. Conclusions

In conclusion, implementing a health promotion program combining students information during school classes, parents and general practitioners information, and offering a free school-based vaccination on Reunion Island, could raise the vaccination coverage and hopefully someday decrease the burden of condylomas, cervical dysplasia and cancer and other HPV-induced cancers. This conclusion is not new, however it underlines the effective strategies to increase the vaccination coverage in regions with a low adherence.

Barriers to students’ participation, should be understood and discarded before expanding the program to the whole Reunion Island, or even to other regions of the world.

## Figures and Tables

**Figure 1 vaccines-10-01923-f001:**
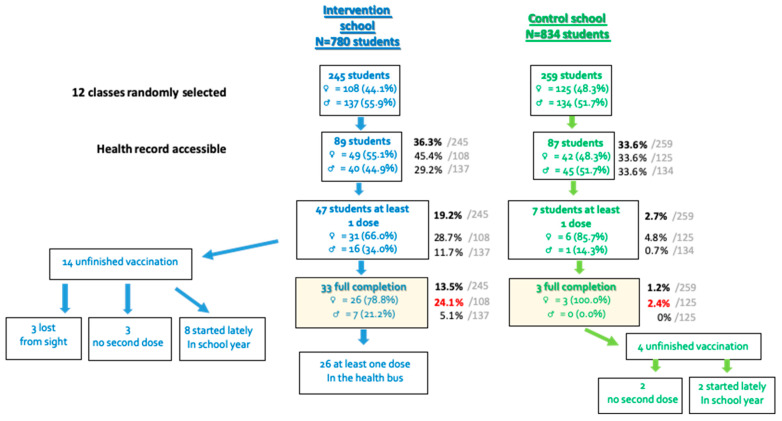
Flow chart. ♀: girls, ♂: boys.

**Table 1 vaccines-10-01923-t001:** Characteristics of children for whom health records were accessible.

		Intervention School	Control School	
		N = 89	N = 87	*p*-Value
		N (%)	N (%)	
Age	9–12 years old	54 (60.7)	55 (63.2)	0.7
	13–16 years old	35 (39.3)	32 (36.8)	
Gender	Female	49 (55.1)	42 (48.3)	0.3
	Male	40 (44.9)	45 (51.7)	
Grade	6th–7th	49 (44.9)	56 (35.6)	0.2
	8th–9th	40 (44.9)	31(64.4)	
Up-to-date vaccinations except HPV vaccination *	yes	50 (41.9)	56 (35.6)	0.4
	no	36 (58.1)	31 (64.4)	

* according to the vaccination schedule in force in France. HPV = human papillomavirus.

## Data Availability

Data supporting reported results are accessible upon request to corresponding author.

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
