# Peer review of "PROM SSCOL—Impact of a Papillomavirus Vaccination Promotion Program in Middle Schools to Raise the Vaccinal Coverage on Reunion Island"

_vaccines, 2022, doi:10.3390/vaccines10111923_

Round 1

Reviewer 1 Report

Even though this manuscript reports on a well-designed study on the effectiveness of information and awareness campaigns on HPV vaccination rates in a very proper and neat manner, it has to be mentioned that the main conclusion of the manuscript is not fully new. I would still recommend the acceptance of the manuscript for publication since it still seems to be important to put emphasis on the topic due to improvable HPV vaccination uptake rates throughout the world. 

Author Response

Response to Reviewer 1 Comments

Point 1: Even though this manuscript reports on a well-designed study on the effectiveness of information and awareness campaigns on HPV vaccination rates in a very proper and neat manner, it has to be mentioned that the main conclusion of the manuscript is not fully new. I would still recommend the acceptance of the manuscript for publication since it still seems to be important to put emphasis on the topic due to improvable HPV vaccination uptake rates throughout the world. 

Response 1: Indeed, the conclusion is not new. We added a sentence in the conclusion to put emphasis on the low HPV vaccination uptake. (p.10, l. 286).

Thank you for your review !

Reviewer 2 Report

The authors present a prospective study in which they demonstrate an increase in HPV vaccination among schoolchildren who received guidance. Despite the weak external validity, the study has the merit of demonstrating that there are effective strategies to increase vaccination coverage in regions with low adherence.

I suggest some corrections before considering acceptance of the manuscript:

Change the last paragraph of the introduction to: "The main objective of the study was to evaluate, in a population of middle school girls and boys in Reunion Island, the impact of a health promotion program, on the proportion of middle school girls and boys who have completed the full HPV vaccination schedule (2 or 3 doses) by the end of the school year. In the same way, add boys in the abstract.

In the Material and methods section, clarify in more detail how the students were selected for the intervention group and explain better in the text how these interventions were, with duration, type of information that was provided and the way they were passed on during classes to boys and girls. In addition, better clarify how parents of children in the intervention group were informed. In addition, they must clearly state in this section if the study was approved by the ethics committee, if the adolescents signed the term, or had to be authorized by their parents to participate, and what was the total duration of the study from acceptance of participation to withdrawal of the vaccination bus.

In the results section, please enter the average age of boys and girls in the two groups and add results on the number of parents of students in the intervention group who participated in the study by phone calls.

In the discussion, I suggest adding a paragraph discussing the low adherence of health professionals on the island to the conference proposed by the researchers, only 7 out of a total of 120. Discuss how this can influence the low vaccination coverage.

The weak external validity of this study and the possibility of boys and girls selected for the intervention group accepting to be vaccinated simply because they help the researchers and not motivated by the information provided in the protocol and applied in the study reinforce the limitations. In addition, talk about the time offered from October to June of the other year, as a limiting problem to be able to offer the two doses with the required interval for students who decided to be vaccinated in the months closer to the end of the study, which can influence the adherence of participants.

Author Response

Response to Reviewer 2 Comments

Point 1: The authors present a prospective study in which they demonstrate an increase in HPV vaccination among schoolchildren who received guidance. Despite the weak external validity, the study has the merit of demonstrating that there are effective strategies to increase vaccination coverage in regions with low adherence.

I suggest some corrections before considering acceptance of the manuscript:

Change the last paragraph of the introduction to: "The main objective of the study was to evaluate, in a population of middle school girls and boys in Reunion Island, the impact of a health promotion program, on the proportion of middle school girls and boys who have completed the full HPV vaccination schedule (2 or 3 doses) by the end of the school year. In the same way, add boys in the abstract.

Response 1: Full protocol is available :

Tran PL, Chirpaz E, Boukerrou M, Bertolotti A. Impact of a Papillomavirus Vaccination Promotion Program in Middle School: Study Protocol for a Cluster Controlled Trial. JMIR Res Protoc. 13 juin 2022;11(6):e35695

Since we refer to this protocol, I suppose that we have to stick to the main objective that was defined in the protocol article ; which was the evaluation of girls only. The rational for the evaluation of these girls, was that when we submitted this protocol to the ethics committee, in October 2019, HPV vaccination in France was recommended for girls only. French recommendations promoted gender-neutral vaccination in December 2019. Since this study was realized soon after recommendation changes, we decided to evaluate girls and boys separately. We added a sentence in the introduction to explain this choice (p. 4, l. 92).

Point 2: In the Material and methods section, clarify in more detail how the students were selected for the intervention group and explain better in the text how these interventions were, with duration, type of information that was provided and the way they were passed on during classes to boys and girls. In addition, better clarify how parents of children in the intervention group were informed. In addition, they must clearly state in this section if the study was approved by the ethics committee, if the adolescents signed the term, or had to be authorized by their parents to participate, and what was the total duration of the study from acceptance of participation to withdrawal of the vaccination bus.

Response 2: The two groups (intervention group and control group) were selected from two middle schools in Southern Reunion Island located in priority education zone (which theoretically concern populations with low socio economic status).

In order to have the most comparable populations in the two arms, we carried out a cluster trial. In each of the schools, we randomly drew three classes in each grade level (6th, 7th, 8th, and 9th grade) to have a balanced number of students in each arm. Thus, 12 classes will be selected for each school. (p.5 l. 103)

Children in the selected classes were asked to bring in their health record on a specific date. On that day, an investigator collected data necessary for the study in the health records (especially vaccination data) for children for whom consent form was signed by the parents. During this time, an information session about anatomy of genital organ, sexually transmitted diseases and vaccination was given in class, lasting approximately 1 hour and adapted to the level of understanding (according to grade and age), in partnership with teachers. Health records were immediately returned to the students (p.5, l. 115).

Each child was given an envelope to take home, with consent form to participate to the study to be signed : by the child him/herself and both parents or holders of parental authority. Since assemblies and parental meetings were forbidden during the COVID-19 pandemic, parents were phoned and oral consent was collected (p.5, l. 121).

This research has received the favorable opinion of the research ethics committee (Comité de Protection des personnes (CPP); ethics committee for the protection of individuals) of Ouest II of Angers (No. 20.05.14.35227; 2020/46) and the authorization of the Agence nationale de la sécurité du médicament (ANSM) (p. 6, l. 161).

Total duration of the study was one school year, from October 2020 to June 2021. Participants or their parents could withdraw their consent at any time.

Point 3: In the results section, please enter the average age of boys and girls in the two groups and add results on the number of parents of students in the intervention group who participated in the study by phone calls.

Response 3: Mean age for girls in intervention group was 12,2 years (CI95% : [11,8 ; 12,6]) vs control group 12,2 years (CI95% : [11,8 ; 12,7]) ; p=0,7. As for boys mean age in intervention group was 12,3 years (CI95% : [11,9 ; 12,7]) vs control group 11,9 years (CI95% : [11,5 ; 12,3]) ; p=0,2 (p.6, l. 164).

Of 245 children, some only had one holder of parental authority. For those who had both parents, we tried to call both parents, but when one of them was not available, we considered one oral consent was sufficient. However, both parents’ written consent was necessary. Among 199 mothers who picked up the phone, 79 gave their oral consent for their child’s vaccination. Among fathers, 39 gave their oral consent for vaccination among 113 who answered the phone. (p.7, l. 180).

Point 4: In the discussion, I suggest adding a paragraph discussing the low adherence of health professionals on the island to the conference proposed by the researchers, only 7 out of a total of 120. Discuss how this can influence the low vaccination coverage.

Response 4: A paragraph was added (p.9, l. 248). This low adherence in the conference could reflect a lack of interest and thus of prescription of HPV vaccination. Yet, as said earlier, patients have a deep trust in their general practitioner and the doctor needs to be convinced himself in order to convince the patient and parents. Improvements are needed to sensitize healthcare professionals.

Point 5: The weak external validity of this study and the possibility of boys and girls selected for the intervention group accepting to be vaccinated simply because they help the researchers and not motivated by the information provided in the protocol and applied in the study reinforce the limitations. In addition, talk about the time offered from October to June of the other year, as a limiting problem to be able to offer the two doses with the required interval for students who decided to be vaccinated in the months closer to the end of the study, which can influence the adherence of participants.

Response 5: These limits were added in the discussion :

“Even though unlikely, students could have accepted to be vaccinated because they wanted to help the researchers and not because they were motivated by the information provided. On the contrary, the wide time offered between first and last vaccination campaign (December to June), could have been a cause of low adherence to second dose. External validity is needed by extending this program throughout Reunion Island. Since World Health Organization seems to say that one vaccinal dose may be sufficient, maybe only one-single vaccination campaign  may be required.” (p.9, l. 271).

Thank you for your review !

Reviewer 3 Report

PROM SSCOL

Unfortunately, the design, assigning one school to condition, does not allow for any firm conclusions from the statistical analyses. Since the unit of assignment (schools) should be the unit of analysis, inferential statistics are problematic in this study.

Even if there were not this fatal design flaw, not enough is known about the intervention to assess findings. What is meant by student and parent information? What was in the practitioner’s letter and call? Perhaps even more importantly, we do not know if merely providing free vaccination is driving the effects? Practitioner endorsement has proven powerful in previous studies. Maybe this endorsement plus availability drives effects?

We also do not know enough about the data collection. What measures were used?

While the findings are promising, the limitations noted in the paper (e.g., low participation rate of 35%) as well as those described above detract from the contribution this paper can make.

Minor point:  While it is technically true to say “it is important to vaccinate before the beginning of sexual life” this needs to be qualified to note the follow up vaccinations after age 12 are also recommended.

Author Response

Response to Reviewer 3 Comments

Point 1: Unfortunately, the design, assigning one school to condition, does not allow for any firm conclusions from the statistical analyses. Since the unit of assignment (schools) should be the unit of analysis, inferential statistics are problematic in this study.

Even if there were not this fatal design flaw, not enough is known about the intervention to assess findings. What is meant by student and parent information? What was in the practitioner’s letter and call?

Response 1: Full protocol is available :

Tran PL, Chirpaz E, Boukerrou M, Bertolotti A. Impact of a Papillomavirus Vaccination Promotion Program in Middle School: Study Protocol for a Cluster Controlled Trial. JMIR Res Protoc. 13 juin 2022;11(6):e35695

Children in the selected classes were asked to bring in their health record on a specific date. On that day, an investigator collected data necessary for the study in the health records (especially vaccination data) for children for whom consent form was signed by the parents. During this time, an information session about anatomy of genital organ, sexually transmitted diseases and vaccination was given in class, lasting approximately 1 hour and adapted to the level of understanding (according to grade and age), in partnership with teachers. Health records were immediately returned to the students.

Each child was given an envelope to take home, with consent form to participate to the study to be signed : by the child him/herself and both parents or holders of parental authority. There was also an information letter, explaining the benefice of HPV vaccination. Since assemblies and parental meetings were forbidden during the COVID-19 pandemic, parents were phoned and oral consent was collected. During phone calls, oral information about HPV vaccination was delivered.

We added supplementary sentences in the Material and methods to describe more precisely what was meant by students and parent information, and the content of practitioner’s letter and call (p.5 l. 115).

Point 2: Perhaps even more importantly, we do not know if merely providing free vaccination is driving the effects? Practitioner endorsement has proven powerful in previous studies. Maybe this endorsement plus availability drives effects?

Response 2: Free vaccination was one of the factor influencing HPV vaccination that we wanted to evaluate. Indeed, in the introduction, we said :

“Moreover, vaccination coverage rates depend on socioeconomic level of the population15, and Reunion Island is one of the poorest French department16. “

  1. Blondel C, Barret AS, Pelat C, Lucas E, Fonteneau L, Lévy-Bruhl D. Influence des facteurs socio-économiques sur la vaccination contre les infections à papillomavirus humain chez les adolescentes en France [Internet]. [cité 11 avr 2022]. Disponible sur: https://www.santepubliquefrance.fr/import/influence-des-facteurs-socio-economiques-sur-la-vaccination-contre-les-infections-a-papillomavirus-humain-chez-les-adolescentes-en-france
  2. Dehon M. Plus de travailleurs pauvres ou modestes exposés à une perte de revenus qu’ailleurs - Insee Analyses Réunion - 56 [Internet]. [cité 11 avr 2022]. Disponible sur: https://www.insee.fr/fr/statistiques/5356765

However, our financing was not sufficient to evaluate each factor eperatly (information on one side and free vaccination at school on the other), because it would have added another group to vaccinate. We also believe that these factors are complementary. A sentence was added in the limitations (discussion, p. 9, l. 269).

Practitioner endorsement is absolutely essential to convince adolescents and their parents. This is the reason why information of general practitioner needs to be reinforced.

Point 3: We also do not know enough about the data collection. What measures were used?

Response 3: Data were collected after consent was signed by parents or holders of parental authority and by students. Data were collected in the form of self-questionnaires (parents and children), evaluating socioeconomical data, and prior knowledge and acceptance of HPV vaccination. Data concerning vaccination status at inclusion and at the end of the study were checked in the health record by the investigation team. Data were collected in paper format and were then entered into an electronic case report form (Ennov Clinical) by a clinical study technician (p.6, l. 143).

Point 4: While the findings are promising, the limitations noted in the paper (e.g., low participation rate of 35%) as well as those described above detract from the contribution this paper can make.

Response 4: The limitations were added in the discussion. We are conscious that this paper’s conclusion is not new, however it underlines effective strategies to increase vaccination coverage in regions with low adherence.

Point 5: Minor point: While it is technically true to say “it is important to vaccinate before the beginning of sexual life” this needs to be qualified to note the follow up vaccinations after age 12 are also recommended.

Response 5: We changed the sentence into “Because HPV is mainly transmitted sexually, it is important to vaccinate at young age.” (p. 4, l. 75)

Thank you for your review.

Round 2

Reviewer 3 Report

Nice job responding to the issues raised. The use of a single school is still problematic but this limitation is noted.